# A Review on Recent Approaches on Molecular Docking Studies of Novel Compounds Targeting Acetylcholinesterase in Alzheimer Disease

**DOI:** 10.3390/molecules28031084

**Published:** 2023-01-21

**Authors:** Stergiani-Chrysovalanti Peitzika, Eleni Pontiki

**Affiliations:** Department of Pharmaceutical Chemistry, School of Pharmacy, Faculty of Health Sciences, Aristotle University of Thessaloniki, 54124 Thessaloniki, Greece

**Keywords:** acetylocholinesterase, Alzheimer’s disease, AChE inhibitors, molecular docking, simulation techniques, peripheral anionic site, catalytic active site

## Abstract

Alzheimer’s disease (AD), a neurodegenerative brain disorder that affects millions of people worldwide, is characterized by memory loss and cognitive decline. Low levels of acetylcholine and abnormal levels of beta-amyloid, T protein aggregation, inflammation, and oxidative stress, have been associated with AD, and therefore, research has been oriented towards the cholinergic system and primarily on acetylcholinesterase (AChE) inhibitors. In this review, we are focusing on the discovery of AChE inhibitors using computer-based modeling and simulation techniques, covering the recent literature from 2018–2022. More specifically, the review discusses the structures of novel, potent acetylcholinesterase inhibitors and their binding mode to AChE, as well as the physicochemical requirements for the design of potential AChE inhibitors.

## 1. Introduction

Alzheimer’s disease (AD), a neurodegenerative brain disorder, increasingly affects millions of people and typically presents as cognitive debility and memory loss [1]. The main symptoms of AD include neurodegeneration, the deterioration of brain functions, apathy, anxiety, delusions, and depression [2,3,4]. The lack of specific treatments for AD highlights the significance of an accurate diagnosis. AD is usually diagnosed with Magnetic Resonance Imaging (MRI) or Positron Emission Tomography (PET). MRI presents images with high-quality contrast of soft tissues and better spatial analysis with no risks for the patient [2], while PET uses radiotracers to visualize, measure, and record physiological changes in metabolism, neurotransmitters, blood flow, etc. [5].

Currently, there is no effective treatment for AD, which may be associated partially with a lack of a clear underlying mechanism. Based on the fact that AD is characterized by the appearance of extracellular amyloid-beta (Aβ) plaques and neurofibrillary tangles in the intracellular environment, gliosis, loss of synapses, and inflammation, a number of hypotheses have been advanced to explain AD [6,7,8,9]. These include (a) the Aβ-amyloid hypothesis, (b) the Aβ-amyloid oligomer hypothesis, (c) the tau hypothesis, (d) the Ca^2+^ dysregulation hypothesis, (e) the presenilin hypothesis, and (f) the lysosome hypothesis. According to the hypotheses associated with Aβ-amyloid there is an overproduction of Aβ-amyloid peptide leading to the amyloid plaques synapto- and neurotoxicity and neurodegeneration [10]. According to the tau hypothesis, there is an abnormal tau (tubulin associated unit) phosphorylation, resulting in the formation of abnormal neurofibrillary structures. Tau proteins in normal cells bind to microtubules and promote their stability and polymerization [11]. The Ca^2+^ dysregulation hypothesis is based on the calcium sensing receptor (CaSR), a member of the family C of G protein-coupled receptors (GPCRs), mediating calcium homeostasis and regulating intracellular signals [12]. It has been found that CaSR dysregulation is associated with inflammation and neurodegenerative disorders such as AD [13,14]. The presenilin (PS) hypothesis is based on inherited mutations in the genes encoding presenilins, the catalytic subunit of γ-secretase, cleaving the amyloid precursor protein (APP) and, thus, contributing to an increased vulnerability of the brain and AD [15]. Finally, the lysosome hypothesis is based on mutations in genes regulating lysosomal pH resulting in impairment of the autophagy–lysosomal pathway [16].

Two types of enzymes are associated with the disease: acetylcholinesterase (AChE) and butyrylcholinesterase (BChE). It has been found that they can accelerate the Aβ peptide assembly into Alzheimer-type aggregates, increasing their neurotoxicity [17]. In patients with AD, low levels of acetylcholine, abnormal levels of β-amyloid, aggregation of T proteins, inflammation, and oxidate stress have been observed [18]. Alzheimer research is focusing on the cholinergic system and mostly on acetylcholinesterase (AChE) inhibitors.

AChE belongs to the carboxylesterase family of enzymes, type-B. Carboxylesterase type B is a family of evolutionarily related proteins [19]. It is a serine hydrolase that catalyzes the breakdown of acetylcholine and other choline esters that function as neurotransmitters [20]. Due to its quick rate of catalysis, AChE, which is produced by muscle, nerve, and hematopoietic cells, is regarded as one of the most effective enzymes. The AChE active site is a 20 Å deep gorge, and three amino acids (Ser203, Glu334, and His447) are forming the catalytic triad. A peripheral binding site is present nearby, beyond Tyr337. AChE inhibitors can interact with both of these sites [21]. Kinetic studies have shown that AChE has two different active sites, the esteratic and the anionic, corresponding to the catalytic machinery and the choline-binding pocket [22]. The esteratic site, where acetylcholine is hydrolyzed to acetate and choline, contains the catalytic triad (Ser203, Glu334, and His447). The serine and histidine residues are present in other serine proteases, but the third amino acid is aspartate. In addition, the catalytic triad in AChE is of opposite chirality compared with other proteases [23]. Mechanistically, after the carboxylate is hydrolyzed to free choline and acyl-enzyme, the latter undergoes a nucleophilic attack by a water molecule and is assisted by the histidine. The result of this reaction is the release of acetic acid and the free enzyme [24].

In the anionic site, the c reaction involves the positive quaternary amine of acetylcholine as well as other cationic substrates and inhibitors. Studies have demonstrated that the anionic site is lipophilic, uncharged, and contains aromatic residues in the active site [22].

Overall, the majority of the human’s AChE isoforms active site consist mainly of the following subsites (Figure 1): the mid-aromatic gorge; the catalytic triad (CT) containing Ser203, Glu334, and His447, as mentioned above; the peripheral anionic site (PAS), composed of Trp286, Tyr124, Asp74, Ser125, and Phe295; the omega loop (OL); the oxyanion hole (OH); the anionic subsite (AS), including Gly448, Glu202, Ile451, Tyr133, and Trp86; and acyl binding pocket [25,26,27]. Based on their mechanism of action, acetylcholinesterase inhibitors are classified as irreversible, reversible, or pseudo-irreversible [28].

## 2. Inhibition

Known acetylcholinesterase inhibitors include Tacrine, Galantamine, Donepezil, Rivastigmine, Physostigmine/Eserine, and Quinazoline (Figure 2). These compounds are often used as reference compounds for the discovery of novel compounds. Various compound classes have been recently investigated as AChE inhibitors, targeting the active site of the enzyme. In the last five years (2018–2022), a number of novel derivatives have been developed and studied for their interaction with AChE. These novel derivatives and the reference compounds, as well as their inhibitory values and docking results, are briefly presented in Appendix A in the Appendix A. In this review, data from recent literature based on molecular modeling, docking, and simulation techniques have been collected. The results of the novel compounds have been classified based on their molecular structure and mechanism of action.

### 2.1. Coumarin Derivatives

Coumarin, or 2H-1-benzopyran-2-one, is present in various natural products and synthetic compounds. It is known for its antioxidant, anticancer, and inhibitory activity against AChE [29,30]. Coumarin is widely used in the pharmaceutical industry, mainly for the inhibition of AChE, as it shows many possible substitution sites. Amin, K.M. et al., carried out a molecular docking study to understand the binding mode of coumarin derivatives in the active site of AChE in comparison with the known drug donepezil (Figure 1). The presence of a benzyloxy moiety played an important role in the binding mode. The most active compounds showed a common predicted binding pattern at AChE’s binding site. The benzyloxy moiety of the novel derivatives seems to occupy the same region as the benzylpiperidine moiety of the inhibitor donepezil and interacts via π-π stacking with Trp86. A water-mediated hydrogen bond is formed between the carbonyl group of coumarin and Asp74, Tyr337, and Tyr341, similarly to the protonated piperidine in donepezil. Additionally, in coumarin derivatives, two additional hydrogen bonds seem to be formed between the oxygen of the benzyloxy moiety and Ser203 and between the carbonyl group of the coumarin core, acting as a hydrogen bond acceptor, and Tyr341 [30].

#### 2.1.1. Coumarin-Pyrazoles Hybrids

Benazzouz-Touami, A. et al., synthesized a series of coumarin-pyrazole hybrids as AChE inhibitors. The two most active compounds bear are (a) a nitro group in the 6 position and (b) an extra benzoyl ring on the coumarin scaffold. The chromenone ring of the compounds interacts with Tyr341 and Trp286 at the entrance of the active site gorge of human AChE in an attractive and noncovalent hydrophobic manner, resulting in π-π stacking and π-Alkyl interactions. Regarding the binding mode of the nitro group, the two oxygen atoms are interacting with hydrogen bonds with the hydroxyl groups of residues Tyr337 and Tyr124. Additionally, the coumarin carbonyl oxygen forms a hydrogen bond with the backbone nitrogen of the Phe295. Additional interactions are observed from the pyrazole moiety, where the -NH moiety interacts with the carbonyl group of Tyr341 and the nitrogen atom of the pyrazole forms a hydrogen bond with the side chain hydroxyl oxygen of Ser293 [31].

The addition of a benzyl group to the chromen-2-one ring reduced the affinity towards the AChE enzyme. Furthermore, there is an absence of hydrogen bond interactions with Tyr124, Tyr337, and Phe295 compared with compound **5** (Figure 2). As a result, the carbonyl oxygen of the chromen-2-one ring interacts with the nitrogen backbone (NH) of Arg296 and the hydroxyl group of the side chain of Ser293, whereas the nitrogen of the pyrazole ring forms a hydrogen bond with the backbone carbonyl oxygen of Phe338. On the contrary, 3-(5-methyl-1H-pyrazol-3-yl)-2H-benzo[g]chromen-2-one, compound **6** (Figure 2), with Tyr341, Phe297, Trp286 and Leu289, residues of the entrapment of the active site gorge, resulting in π-π stacking, π-π T-shaped, and π-Alkyl interactions. As shown by the crystal structure of human AChE, similar interactions were developed with residues Tyr124, Tyr337, Phe295, Tyr341, Trp86, and Asp74. These findings suggested that reducing the favorable contacts with the catalytic residues of the AChE, and particularly with Tyr337, which has been demonstrated to be crucial for the inhibition of the human enzyme, may be the cause of the decreased affinity of benzyl derivative compared to 6-NO_2_- derivative [31].

#### 2.1.2. Thiophene Chalcone-Based Coumarin Analogues

In 2022, Hasan, A.H. et al., reported a series of novel thiophene chalcone-based coumarin analogues. The coumarin moiety (compound **7**, Figure 3) was located in the catalytic anionic site (CAS) of the enzyme and was interacting with Gly121 and Gly122 via two hydrogen bonds at the carbonyl oxygen. The Oxygal group of the chalcone moiety and the amino acids Tyr72 and Trp286 also seemed to form two additional hydrogen bonds at the peripheral anionic site (PAS) region. The phenyl ring of coumarin formed two π-T-shaped interactions with Tyr337 and Tyr341. There were also two additional π-T-shaped contacts between His447 and the methyl substituent, as well as between the chalcone core’s thiophene ring and the His287. In contrast, Trp286 and another phenyl ring of chalcone were able to form two π-π stacking contacts. The coumarin core’s methyl group interacted with Tyr337 and Trp86 through π-alkyl interactions. The coumarin pyrone ring and the amino acid His447 interacted via electrostatic (π-cation) and π-alkyl interactions. The sulfur atom and His287 amino acid residue interacted via the sulfur bond [32].

### 2.2. Chalcone Derivatives

Chalcones, or α,β-unsaturated ketones or open-chain flavonoids, are bioactive compounds, which can interact with a variety of enzyme targets because of the three rotatable bonds in their structure and their capacity to act as the Michael acceptors [33]. Consequently, a variety of pharmacologically active substances containing the chalcone group have anti-inflammatory, anticancer, antidiabetic, and anti-AD properties [34]. Malik, Y.A., and co-workers synthesized a series of chalcones and studied them for their binding mode to acetylcholinesterase. Τhe most active compound was the (*E*)-1-(4-chlolorophenyl)-l3)-3-phenylprop-2-en-1-one. The carbonyl group (-C=O), the α-β unsaturation (double bond), the aryl rings A and B, and the substitutions on the aryl ring are the major functionalities found in the chalcone scaffold (R and R_1_). The molecular interactions displayed by these groups were studied using molecular docking studies, and their binding energies (docking scores) were computed and correlated with the in vitro AChE inhibition data. It was hypothesized that the carbonyl functionality would play a role in the formation of hydrogen bonds with the His and Ser residues found in the protein’s binding site. The aromatic amino acids Phe, Trp, and Tyr interacted with the aryl rings A and B in a π-π interaction. Additionally, it was discovered that these aryl rings play a role in the π-cation interaction between the aryl ring and the quaternary nitrogen atom (ammonium cation) found in amino acids like histidine. The binding affinity of compounds was also significantly impacted by substitutions on the aromatic ring. The interaction between the -OH group in the tyrosine residue and the Cl atom in compound *(E)*-1-(4-chlolorophenyl)-l3)-3-phenylprop-2-en-1-one (compound **8**, Figure 4) was important. In compounds *(E)-*1-(4-ethylphenyl)-3-(4-nitrophenyl)prop-2-en-1-one (compound **9**, Figure 4) and *(E)*-1-(4-bromophenyl)-3-(4-nitrophenyl)prop-2-en-1-one (compound **12**, Figure 4), the oxygen of the nitro group formed hydrogen bonds with the amino acids Glu, Asp, and Tyr. Similar to the second compound, *(E)-*1-(4-chlorophenyl)-3-(4-methoxyphenyl)prop-2-en-1-one (compound **10**, Figure 4) the methoxy group’s oxygen atoms participated in hydrogen bonding with glycine, making the affinity even stronger. The substitution of *(E*)-1-(4-bromophenyl)-3-(4-hydroxyphenyl)prop-2-en-1-one (compound **11**, Figure 4) with a hydroxyl group showed increased binding affinity *via* the formation of hydrogen bonds with the serine and glutamine residues. The chalcone scaffold and various amino acid residues present at the binding pocket of the enzyme, including Gly, Tyr, Ser, Phe, Trp, Leu, and Ile, were expected to form hydrophobic interactions (Figure 5, Figure 6, Figure 7, Figure 8 and Figure 9) [35].

### 2.3. 4-Aminobenzohydrazide Derivative

Almaz, Z. et al. [36] investigated the inhibitory effects of several mono- or di-substituted 4-aminobenzohydrazide derivatives against AChE, exploiting the biological activities and the pharmacological effects of the benzohydrazide moiety. The most active derivatives were docked into the AChE active sites. The 4-amino-3-bromo-5-fluorobenzohydrazide (compound **13**, Figure 10), being the most active derivative within this series, formed van der Waals interactions with the residues Tyr115, Trp113, Ile447, Gly444, Tyr129, Ser 199, Gly117, Gly118, and Tyr120. The hydrazide moiety interacted with Gly116 in a typical hydrogen bond, Trp82 in a cation-cation interaction, Gly444 in a carbon-hydrogen bond, a salt bridge, and Glu198 residue in a charged interaction. The Phe293, His443, and Phe334 residues established -alkyl interactions with the Br substituent of its benzene ring. Furthermore, the compound’s benzene ring interacted in a π-T shape with the Trp82 and Tyr333 residues [36].

### 2.4. Aminobenzoic Acid Derivatives

Iftikhar’s K. et al., focused on the study of aminobenzoic acid derivatives bearing a carboxyl group with an ionizable proton, which could enhance their potency by reducing their lipophilic nature. The most potent compound was 3,3′-(Isophthaloylbis(azanediyl))dibenzoic acid (compound **14**, Figure 11). This compound formed several π-π interactions with residue Tyr337 and Trp86 of the AChE’s anionic binding site. No hydrogen bonds were observed with the catalytic triad’s amino acid residues. However, the carboxylic group of the aminobenzoic acid formed a hydrogen bond with the Asp74 and Tyr341 residues of the peripheral binding site. Additional hydrogen bonds between this compound and the residues Tyr72 and Thr83 were observed [37].

### 2.5. Pyrrolidine Derivatives

A group of N-benzylpyrrolidine-based compounds were designed and synthesized by El Khatabi, K. et al., and biologically investigated as potential acetylcholinesterase inhibitors. In this current work, new N-benzylpyrrolidine derivatives were particularly suggested using ligand-based and structure-based drug design investigations, among which 3D-QSAR techniques provided theoretical guidance to elucidate the structure-activity relationships of compounds. Molecular docking studies were also used to investigate the precise binding properties and dynamic behavior of the chosen inhibitors to the target protein. The most active compound (compound **15**, Figure 12) was able to produce important interactions such as hydrophobic, hydrogen, and halogen bonds. Residues Tyr121 and Asp72 c formed two hydrogen bonds with the NH linker that connects the right phenyl moiety to the pyrroline ring in the molecule. The inhibitory activity benefits from these interactions. The phenyl moiety on Trp84’s left side forms a π-π stacking that is advantageous to the binding. Additionally, the neighbouring Phe290 residue positioned close to R_1_ substituent and the OCF(C_2_H_5_)_2_ of the core pyrrolidine ring produce a π-sigma effect, indicating that the hydrophobic group at the R_1_ substituent is responsible for compound’s improved biological activity [38].

### 2.6. Pyrrole Derivatives

It is well established that the *N*-heterocyclic moieties correlate with a variety of biological and therapeutic effects. Pyrrole derivatives present a wide number of biological activities, including antioxidant, anti-inflammatory, analgesic, antidiabetic, and anticancer agents. Additionally, the pyrrole ring is a crucial core component of numerous biological molecules, including alkaloids, heme, vitamin B12, bile pigments, and chlorophyll. The creation of a novel and straightforward synthetic approach for the effective synthesis of novel polysubstituted pyrroles will be a helpful and alluring challenge because of their therapeutic capabilities [39].

For this reason, Pourtaher, H. and collaborators [40] designed and synthesized a series of pyrrole derivatives that were further evaluated for AChE inhibition. They used computational docking studies to simulate 2-(2-(4-(chlorophenyl))-[1-(4-hydroxyphenyl)]-[5-(methylthio)-[4-nitro]-[1H]-pyrrol-3yl]]-[2-cyanoacetamide] (compound **16**, Figure 13) on AChE’s binding site. In CAS, the nitro and acetamide groups of the compound seem to be important. The acetamide group formed three hydrogen bonds with Tyr124 and Trp86 (anionic subsites), while the nitro group interacted with Trp86. 4-OH-benzyl is oriented toward PAS on the other side of the molecule, exhibiting two hydrogen-bound interactions with Phe295 and Tyr341. Additionally, due to the phenyl-ring π staking interactions that are formed with Tyr341. The Thr75 and para chlorophenyl moiety participated in shydrophobic interactions (Figure 14). These interactions supported the biological assessment findings and attested to the high potency of this derivative [40].

### 2.7. Pyrazole Derivatives

Due to their numerous pharmacological properties, including anti-inflammatory, analgesic, anticancer, and antibacterial effects, pyrazole and its derivatives are recognized as a significant class in the field of medicinal chemistry. From a synthetic perspective, numerous synthetic routes could be used to produce a huge number of derivatives, providing a large number of molecules that could be tested for specific biological activity. For instance, acetylcholinesterase (AChE) was inhibited by the tacrine (known drug) functionalized with a pyrazolyl pattern at the nano-molar concentration. In this regard, the utilization of pyrazole-based systems as novel drugs for AChE inhibition is particularly intriguing [41]. Cetin, A., and co-workers focused on pyrazole derivatives with the ultimate goal of inhibiting AChE. The synthesized compounds had good docking scores and proved to be stabilized at the active pocket by hydrogen bonding, π-stacking, and hydrophobic interactions. The docking scores with the relevant target enzymes and binding constants were correlated. *N*,1-Diphenyl-3-(thiophen-2-yl)-1*H*-pyrazole-5-carboxamide (compound **17**, Figure 15) showed a high affinity for AChE binding and was discovered to engage hydrophobically with Val294, whereas a hydrogen bond was formed with His447, as well as with cations, anions, and stacking interactions. To increase the binding affinity of this compound, other aromatic amines can be added [42].

### 2.8. Imidazole Derivatives

It is generally known that substituted imidazole moieties have a variety of medicinal effects. Imidazoles are potent farnesyltransferase inhibitors, potent kinase inhibitors, and active 5-lipoxygenase inhibitors [43,44,45]. Based on that, Pervaiz S. and her research team synthesized and studied 2,4,5-trisubstituted imidazoles with possible potent inhibitory activity against acetylcholesterase. Τhe promising results were validated by Molecular Docking Studies. Docking interactions of the novel derivatives resulted from the appropriate substitution of the imidazole core placed into a residue pocket created by Thr121, Phe290, Phe331, Phe330, Typ84, Tyr334, Asp72, Tyr70, Trp270. The main interaction in this series is between the amino group and the oxygen atom of the residue Thr121 through an H-bond. Furthermore, the 2-(4-Methoxyphenyl)-4,5-diphenyl-1H-imidazole’s (compound **18**, Figure 16) N-atom creates an H-bond with the oxygen atom of the residue Thr121 of AChE. Last but not least, in some imidazole inhibitors, the amino group forms an H-bond with the oxygen atom of the residue Ser122, demonstrating a different primary configuration [46].

### 2.9. 1,3,4 Oxadiazole Derivatives

The 1,3,4-oxadiazole is a bioisosteric pentacyclic molecule interacting with particular domains on enzyme active sites. Small changes in the 1,3,4-oxadiazole structure have significant impact on biological activities such as the inhibition of acetylcholinesterase [47].

In an interesting study by Tariq, S. et al., it was attempted to synthesize several 1,3,4-oxadiazole derivatives substituted by thio-alkyl. The results revealed a few compounds that could potentially act as “lead” molecules against the intended enzymes. In Figure 17, three more active compounds are shown. Variable docking results are primarily due to the different active functional groups of these inhibitors. The novel derivatives are placed into AChE residue pockets formed by Ser125, Thr83, Asp74, Tyr124, Tyr133, Tyr337, Tyr449, His447, and Gly448. It seems that the three more potent compounds of these series present differences in their binding mode eg. compound **19** (Figure 17) formed π-π interactions between its aromatic rings and Tyr337 and Trp86 and a strong H-bond with its hydroxy group and His447. While compound **20’s** (Figure 17) benzene ring displayed π-π interactions with Trp86 and Tyr337, its hydroxy group developed a H-bond with Thr83. Finally, compound **21** (Figure 17), bearing another hydroxy group at the para position, presented the best interaction through H-bonding and insertion into the protein pockets at Tyr133 and Glh202. Additionally, π-π interaction of the oxadiazole ring with Tyr124 and benzene ring interactions with Trp86 were formed [48].

### 2.10. Lophine Derivatives

The structure of lophine, or 2,4,5-triphenyl-1H-imidazole, consists of three phenyl rings attached to the imidazole core that are not coplanar. The dihedral angles between the phenyl ring planes in the 2-, 4-, and 5-positions of the imidazole ring are 21.4°, 24.7°, and 39°, respectively [49]. In Lopes’s B. and co-workers’ study, lophine-based hybrids connected with tacrine by alkyl linkers are presented, showing considerable AChE inhibitory activity [50].

The only potent compound was *N1*-(1-O-Methyl-5-deoxy-2,3-O-isopropylidene-β-D-ribofuranoside)-5-(N7-(2,4,5-triphenyl-1H-imidazol-1-yl)heptan-1-amine), compound **22** (Figure 18). This compound has the ribose moiety packed between Trp84 and Phe330, in accordance with the glide binding modes, whereas the lophine moiety has one phenyl group packed between Tyr70 and Trp279 from PAS. The linker has a “bend” in the compound binding mode, allowing a second phenyl ring to connect with several aromatic residues in a hydrophobic pocket in the centre of the binding cavity, which is primarily made up by Try121, Phe290, Phe331, and Phe330 [50].

### 2.11. 1,2,4-Triazole Derivatives

Triazoles are a subclass of heterocyclic compounds and are of particular interest in drug discovery. A variety of enzymes and receptors in biological systems can readily interact with triazoles and their derivatives through coordination or hydrogen bonds, ion-dipole, cation-π, π-stacking, hydrophobic effect, and van der Waals forces. This is due to the three nitrogen atoms present in triazoles and the accompanying rich electron system [51,52]. Triazole nucleus’ broad range of therapeutic uses are also attributed to its low toxicity, few side effects, excellent bioavailability, favorable pharmacokinetic characteristics, and drug-targeting capabilities [53]. The 1,2,4-triazole nucleus-derived medications, such as etizolam, alprazolam, and voriconazole, are used as muscle relaxants, antifungal medications, and herbicides to treat inflammation, respectively. Numerous biological properties, including fungicidal, antibacterial, anticonvulsant, anticancer, and antioxidant, are also demonstrated by 1,2,4-triazoles with mercapto substitution on ring systems [54,55,56]. In addition, chemicals derived from triazoles are effective inhibitors of cholinesterase enzymes [57].

Triazoles have a broad range of biological effects, which inspired Riaz’s, N. et al. [21] to synthesize more analogs and further investigate their cholinesterase inhibition capabilities. In Figure 19, four more active compounds are presented. The docked conformation of the first compound’s (compound **23,** Figure 19) binding site interactions reveals a number of non-bonded interactions that are in charge and stabilize the enzyme-inhibitor complex. While Phe295 is also within the hydrogen bond range of the sulfur atom, one of the triazole ring’s nitrogen atoms was forming hydrogen bonds with Arg296 and Phe295. Tyr124 and the NH of the amide group were forming a hydrogen bond. In particular, π-alkyl interactions between the triazole ring and Val 294 as well as between the N-ethyl group and Trp 286 and Tyr 341 were noted. The interaction between Tyr337 and the trichloro phenyl ring was a π-π T-shaped interaction [21]. Docking tests for the second most active inhibitor (compound **24,** Figure 19) showed that Thr75 and the carboxylic acid moiety formed a hydrogen bond. While one of the triazole ring’s nitrogens formed a hydrogen bond with Tyr124, the NH of the amide group formed one with Asp74. Tyr124 and Tyr337 were engaged in π-π T-shaped interactions with the triazole ring. The aromatic Tyr341, Tyr337, and Phe338 rings were interacting with the N-ethyl group through π-alkyl interactions. Leu76 and the nitro phenyl ring were also interacting via an alkyl group [21]. Compound **25** (Figure 19) was discovered to bind a little bit more distantly from the cavity’s outside opening, along the peripheral binding site. It was discovered that the triazole ring nitrogen atoms formed hydrogen bonds with Arg 296 whereas the sulfur atom formed hydrogen bonds with Ser 293 in the compound. Trp286 and the NH of the amide group were hydrogen bonded. The nitro phenyl ring’s methyl group formed a π-alkyl interaction with His287. Additionally, the triazole ring contributed to the π- alkyl contact with Arg296. Trp286 and the N-ethyl group were interacting via an π-alkyl group. Tyr341 and Phe296 were interacting with the chloro-phenyl ring in stacked and π-π T-shaped ways, respectively [21]. The most effective AChE inhibitor was compound **26** (Figure 19). While the nitrogen atoms of the triazole ring formed hydrogen bonds with Gly116, Gly117, and Ala199, the amide NH formed a hydrogen bond with His438. Trp231 and Phe329 were interacting in a π-π T-shaped with the chloro phenyl group. The neighboring fluoro phenyl ring and the N-phenyl ring of the same molecule made an intramolecular π-π stacked interaction, while Phe329 was also making a π-π stacked interaction with the N-phenyl ring. Additionally, Trp82 and the fluoro-phenyl ring of **26** were interacting in a stacking fashion. Additionally, a π- sulfur interaction between the S atom and Trp82 was described [21].

### 2.12. Isoniazid Derivatives

Isoniazid is a well-known anti-tuberculosis medication, and the hydrazones that are produced from it are well-known iron chelators. Some chelators, such as clioquinol, are referred to be MPAC because it is believed that they trigger metal release from peptides and the restart of the clearance of β-amyloid [58]. Clioquinol use, however, results in unfavorable side effects [59]. A MPAC has recently been an isonicotinoyl hydrazone of 8-hydroxyquinoline-2-carboxaldehyde. The chemical class of acyl-hydrazones is in fact a potential category of MPACs for the treatment of Alzheimer’s disease, as the same research team affirmed in a subsequent article [60]. Τhis prompted Santos’s D.C. researchers to synthesize hybrid acylhydrazones as isoniazid derivatives. π stacking of the 3-chromonyl nucleus with residues Tyr341, Phe297, and Trp286 (PAS), as well as hydrogen bonding involving the ether oxygen of the 3-chromonyl nucleus and the oxygen atom of the carbonyl group of the acylhydrazone moiety with Ser293 and Phe295 are important interactions of the most active (compound **27**, Figure 20) exhibited with AChE. The peripheral anionic site (PAS) (Tyr74, Tyr124, Trp286 and Tyr341), the catalytic site (Ser203, His447 and Glu334), and the anionic site are all present in the active site of AChE. (Trp86, Tyr133, Tyr337, and Phe338) [61]. In a recent study, the virtual screening of many compounds with anticholinesterase activity revealed that Ser203, Gly122, Tyr124, Tyr72, and Phe337 hydrogen bonds are the most frequently found interactions, as well as π-stacking interactions with the indole ring of Trp286 [62].

### 2.13. Piperazine Derivatives

The piperazine moiety can bind to several receptors with great affinity [63]. Numerous biological activity studies have been conducted, including those on antibacterial [64], antiviral [65], and anticancer properties [66]. Additionally, piperazine-containing compounds are appropriate and promising candidates for AD therapy medications because there are numerous reports of piperazine derivatives inhibiting acetylcholinesterase in the literature [67]. In their study, Sari, S. and Yilmaz, M. [68] analyze the AChE inhibition for piperazine derivatives compared with the effect of donepezil on AChE. The most active derivatives of piperazine are presented in Figure 21. The aromatic groups of His447 and Trp86 interact with by the donepezil N-benzyl moiety through π-π interactions. Moreover, the aromatic moieties of Tyr341, Tyr337, and Phe338 interact with the piperidine ring through π-alkyl and π-π interactions. Additionally, Phe295 and carbonyl oxygen interact as usual through a C-H bond. Through π-π and π-σ bond interactions, Trp286 communicates with the benzene and methoxy groups, respectively. Similar to this, Tyr341 establishes π-π and π-σ interactions with the CH_2_ group and benzene, respectively. It can be demonstrated by examining the interactions between the top docking motif of the first drug and AChE that dimedone methyl interact through π-alkyl bonds with the aromatic groups of His447 at the catalytic triad and Trp86 at the anionic site, similarly to donepezil. Furthermore, the methyl and piperazine rings of dihydrofuran engage via π-alkyl interactions with Tyr341 and Tyr124 at the peripheral anionic site of AChE. Last but not least, the aromatic furan ring interacts (π-π interaction) with Tyr341 [68]. By examining the interactions of the piperazine-dihydrofuran compound **29** (Figure 21) carrying a carbethoxy group, it was found that the amide carbonyl reacts with Phe295 in a way that is comparable to the donepezil carbonyl moiety. Similarly, the dihydrofuran ring’s methyl interacts with the Trp286, Tyr124, and Tyr341 residues in the outer anionic site via π-alkyl linkages. The other methyl group on dihydrofuran interacts with Phe297 and Phe338 at the acyl pocket via π-alkyl interactions, while its ester group interacts with His287 using a typical C-H contact [68]. For derivative **30** (Figure 21) being the most effective inhibitor, ligand-protein interactions showed that Trp286 interacts with both methyl moieties via π-σ and π-alkyl interactions. Additionally, Tyr341, Tyr124, Phe297, and Phe338—form -alky interactions with two dihydrofuran methyls. In addition, the piperazine ring interacts in a way that is comparable to the piperidine ring of donepezil with Phe338 and Tyr337. Furthermore, a π-π interaction between the furan ring and Trp86 occurs. The most effective inhibitors in this work exhibit similar modes of action to the standard medicine donepezil when ligand-protein interactions are considered, and the docking results support the in vitro experimental findings [68].

### 2.14. Pyrimidire Derivatives

Due to the wide range of biological functions that pyrimidine and its numerous analogs exhibit, they have attracted a lot of interest. Pyrimidine is a significant family of heterocyclic compounds with medical benefits as well as a fundamental molecule in the structures of DNA and RNA [69]. In his research, Duran, H.E., composed and studied some simple pyrimidine derivatives. Τhe most active derivative against acetylcholinesterase is 4-amino-5,6-dichloropyrimidine (compound **31**, Figure 22), which forms an H-bond with the pyrimidine ring’s NH_2_ groups and a His447. While the hydrophobic interactions with hydrophobic amino acid residues like Tyr337, Trp439, Pro446, and Tyr449 are the primary elements in stabilizing this derivative, the π-stacking interaction of the amino-5,6-dichloropyrimidine with Trp864 seems to be important. The findings highlighted the NH_2_ moiety’s selectivity in drug development [70].

### 2.15. Pyrazole-Pyridazine Derivatives

In their research, Taslimi, P., and co-workers [71] synthesized compounds consisting of two different nitrogen heterocyclic rings, pyrazole-pyridazine derivatives. From their study, the most active compound was **32** (Figure 23). The docking results showed that practically all of the pyrazole [3,4-d]pyridazine derivatives had strong binding affinities to AChE. Pyridazine is correctly positioned within the receptor’s active catalytic pocket, while created a hydrogen bond with the residues Phe295 and Arg296 as well as a water molecule being constitutional. In addition, a π-stacking interaction was observed between the novel derivative and the Trp286 amino acid residue. Additionally, the compound was surrounded by a number of hydrophobic amino acid residues, including Tyr72, Val294, Leu289, Phe195, Tyr337, Phe338, Trp286, Phe297, and Tyr341 residues [71].

### 2.16. 1,3,5-Triazine Derivatives

With a wide range of biological uses, including antibacterial, antiviral, diuretic, and, more recently, anti-cancer medicines, the 1,3,5-triazine (s-triazine) scaffold has an intriguing pharmacological profile [72,73]. Sulphonamides with a 1,3,5-triazine moiety have recently begun to be researched as effective and specific AChE inhibitors. The peripheral anionic site (PAS) of AChE can interact with the benzimidazole moiety [74]. The formation of AChE-amyloid complexes, which are more neurotoxic than β-amyloid alone, is reportedly stimulated by the PAS of AChE. AChE inhibitors have the ability to interact with the anti-aggregating PAS [75]. Lolaka, N. et al. [60], in 2020, supported the hypothesis that Tyr124 and Leu289 of the AChE binding site form H-bonds with the sulfonamide moieties of the synthesized 1,3,5-triazines derivatives (compound **33**, Figure 24). Additionally, the carbonyl moiety and amino moiety displayed possible H-bonds with His287 and Ser293 in the cavity. Finally, there was π-π stacking between the benzene ring and the Trp286 residue [76].

In another study, Wu, W. L. et al., synthesized 1,3,5-triazines derivatives and compared them with donepezil AChE inhibition. Figure 25 lists the compounds that showed the best interaction with AChE. The most effective AChE-inhibitor, compound **34** (Figure 25), interacted both with the CAS and PAS of AChE in a way that was comparable with donepezil. Through stacking, the compound’s benzimidazole nucleus may interact with Trp86 and Tyr337 of CAS. PAS residue Trp286 in stacking form with the 1,3,5-triazine group. Additionally, the piperazine and benzimidazole nuclei of the compound’s nitrogen atoms are likely to interact with Tyr124 of the CAS through hydrogen bonding. Through the π-cation, the nitrogen atom on the benzimidazole nucleus also communicates with Trp86. All three derivatives seem to be related to AChE in a comparable way. These derivatives therefore have two binding sites for AChE inhibitors. Interestingly, while sharing R groups with compounds **35** and **36**, compounds **37** and **38** interact with AChE differently. This could be attributed to the differences in their benzimidazole groups. Although in distinct ways, compounds **36** and **37** may also interact with the PAS and CAS of AChE. The 1,3,5-triazine group forms multiple stacks with residues Phe338 and His447 of CAS, while the benzimidazole nucleus of the two molecules forms multiple stacking interactions with residue Trp286 of PAS [77].

### 2.17. Pyrrolizine Derivatives

A hydrophobic aromatic moiety, which is analogous to the indanone in donepezil, a nitrogen-containing midgorge binding moiety, and a terminal aryl moiety are the key pharmacophoric features of donepezil-like AChE inhibitors. For this reason, El-Sayed, N.A. et al. [25] designed and synthesized a new series of pyrrolizine derivatives that satisfy the primary pharmacophoric requirements for AChE inhibition. Using Molecular Operating Environment software, a molecular docking analysis was carried out to investigate the potential binding mode of compounds **39** and **40** (Figure 26) within the AChE active site, supporting their efficacy. Three subsites make up the majority of the AChE active site: a peripheral anionic site (PAS), which contains Trp286, Tyr124, Asp74, and Phe295 residues; a mid-aromatic gorge; and a catalytic active site (CAS), which includes Trp86, Glu202, Tyr337, and Gly448 residues. A closer look of compound **39** and compound **40**’s top docking poses revealed that the pyrrolizine heads in both compounds are involved in a significant π-π stacking with the aromatic residue of Trp86 in CAS. Additionally, an H-bond interaction was observed between Glu202 and either the pyrimidinone NH in compound **40** or the carboxamide NH2 in compound **39**. Moreover, an H-bond interaction was developed between Glu202 in the CAS and either the carboxamide NH2 in compound **39** or the pyrimidinone NH in compound **40**. This H-bond corroborated their idea that adding an amide fragment to the pyrrolizine moiety would help to improve the binding and AChE inhibitory activity and may play a significant role in stabilizing the ligands inside the CAS. Additionally, the first compound’ s amide carbonyl group in the linker created an H-bond interaction with Tyr124, whereas for the second compound, the same group formed a water-mediated H-bond with Tyr337 and Tyr341. Both compounds have a protonated nitrogen atom that interacts with Tyr341 by a cation-π interaction and an H-bond with Tyr124, whereas the benzyl moiety was involved in a π-π interaction with Trp286 [25].

### 2.18. Quinoline Derivatives

In their study, Zhu, J., and his co-workers [78] synthesized 4-N-phenylaminoquinoline derivatives and investigated their effects on AChE inhibition. The most active compound’s (compound **41**, Figure 27) N-methylbenzylamine fragment interacted with the CAS via a stacking interaction with Trp84. The 4-N-phenyl ring of the compound also showed a π-alkyl interaction with the sec-butyl moiety of Ile287, which was close to the acyl binding site Phe288. The ortho-chloro group at the 4-N-phenyl ring established a π-alkyl interaction with Trp279 and an alkyl interaction with Leu282. The nitro group formed π-cation and π-anion interactions with Trp279. Finally, the compound interacted with the majority of the amino acid residues in the PAS [78].

### 2.19. Oxindole Derivatives

Oxindole-based substances are thought to be an intriguing scaffold with considerable cholinesterase and VEGFR-2 inhibitory activities. The prevalence of cognitive deficits in individuals with hippocampus-dependent memory and executive functions connected to the frontal lobes that are often affected has been confirmed by neuropsychological studies. Therefore, it is thought to be important to find new medications that are both effective against Alzheimer’s disease and cancer [79,80,81]. In 2022, Srour, A.M., et al. [82] published a very interesting study about oxindole derivatives, which are both effective against Alzheimer’s disease and breast cancer. Regarding AChE inhibition, Trp286, an important amino acid, established arene-arene contacts with the π-tolyl ring and arene-cation interactions with the indoline moiety, leading to the high potency of compounds **42**, **43**, and **44** (Figure 28). Additionally, an arene-cation interaction was observed with compound **43**’s scaffold of 4-methylbenzylidene. Derivative **44**’s 4-methoxybenzylidene moiety was involved in additional arene interactions. The H-bond acceptor in compound **42** between pyrrolidine-N and the sidechain of Tyr72, the H-bond donor in compound **43** between the allyl group and the backbone of Ser293, and the arene-cation interaction in compound **44** between the indoline ring and Glu292 all contributed to the strong binding properties of the previous three compounds with AChE. The analog **45** (Figure 28) was, however, positioned into the AChE binding site via two arene-arene interactions, one between the thiophene ring and Trp86 and the other between indoline and Tyr341 [82].

### 2.20. Quinoxaline Derivatives

Suwanhom, P., and his research team [83] developed novel acetylcholinesterase inhibitors based on quinoxaline scaffolds, using tacrine as a lead drug. Compound **46** (Figure 29) demonstrated hydrogen bonding between the quinoxaline ring and Asp74 and Arg296 as well as π-π interaction with Trp286 and Phe338. For compound **47** (Figure 29), it was observed that the pyrazine ring of quinoxaline bound to PAS through a π-π interaction with Trp286 and Phe338. A hydrogen c bond was observed between Arg296 and the nitrogen atom of the pyrazine ring. A hydrogen bond was formed between the amino group and Ser293 and a hydrophobic contact between the benzyl group and Tyr337. Finally, compound **48** (Figure 29) formed a π-π interaction with Trp286 and Phe338 by utilizing the quinoxaline ring. Furthermore, the methyl group and Ser293 indicated a hydrogen bond, and the nitrogen atom of the pyrazine ring demonstrated a hydrogen connection with Arg296. Finally, the amino group and Asp74 engaged in a hydrogen bonding interaction. In conclusion, the docking prediction showed that compounds tended to prefer the PAS site of the AChE based on the hydrogen bond of the amino group and the π-π stacking of the quinoxaline (Figure 30) [83].

### 2.21. Sulfonate with Aryl α-Hydroxyphsphonate Group

Organic compounds’ heteroatoms significantly influence their chemical reactivity and biological activity. Compounds having aryl sulphonate moieties have drawn a lot of attention over the past years due to their wide range of biological actions, including anti-HIV-1, anti-HIV-2, antineoplastic, and anti-cancer [84]. Phosphonates are a significant subclass of hetero chemicals. Due to their biological activity and value as synthetic precursors for the synthesis of related organophosphonates such as aminophosphonates, alkoxy- and acyloxy-phosphonates, keto-phosphonates, and halophosphonates, hydroxyphosphonates have gained a lot of attention [85,86]. In addition, bis-hydroxyphosphonates make good diol monomers for the creation of phosphorus-containing polymers with flame-retardant qualities [87]. In their study Şahin, I. and co-workers [71], synthesized and developed molecular docking studies about sulfonate compounds with aryl α-Hydroxyphsphonate group. The most active compound (compound **49**, Figure 31) forms hydrogen bonds with Tyr341 in the PAS and Phe295 in the acyl binding pocket, and extra Ser293 and Arg296 residues form to support the bond. Additionally, the compound engaged in a π-π interaction with the Trp286 residue in the peripheral active site, which is surrounded by the amino acid residues Tyr 70, Asp 72, Tyr 121, Trp 279, and Tyr 334. PAS is a protein that is found at the entrance to the active site and is in charge of several crucial functions, including the interaction with β-amyloid. In the acyl binding pocket of AChE, it restricts the freedom of the substrates bound to the phenyl rings of Phe295 and Phe297 and promotes the catalysis of the substrate with the shortest acyl group, such as acetylcholine [87].

### 2.22. Carbazole Based α-Aminophosphonate Derivatives

There are numerous accounts of the multifunctional carbazole moiety being utilized to treat AD. For instance, Choubdar, N. et al., developed new classes of carbazoles [88], Shi, D.H. et al., synthesized carbazole-coumarin hybrids [89], Bachurin, S.O. et al., synthesized novel conjugates of aminoadamantanes with carbazole derivatives [90], and Zhang, X. et al., developed novel multifunctional carbazole-amines [91]. Shaikh, S., and his research team proposed to develop carbazole-based a-aminophosphonates for the treatment of AD, motivated by the biological importance of carbazole and a-aminophosphonates indicated above. The compound diethyl (((9-ethyl-9H-carbazol-3-yl)amino)(3-hydroxyphenyl)methyl)phosphonate (compound **50**, Figure 32 has three π-π stacking interactions with Trp84 (anionic substrate), Phe331 (anionic substrate), and Tyr334 (PAS) residues and two hydrogen bonding interactions with Tyr121 (PAS) and Arg289 residues, according to molecular docking studies [92].

## 3. Conclusions

The application of a wide range of modeling and computational techniques has revealed a strong interest in acetylcholinesterase structure, function, and inhibition, due to its linkage with Alzheimer’s disease, for which no specific cure has been found. This review describes how inhibitors and substrates find and enter the gorges of AChE, selecting data from recent literature from 2018 through 2022. A wide variety of compounds, such as coumarins, chalcones, pyrrolidines, pyrroles, triazoles, quinolines, and more, have shown significant inhibitory activity against acetylcholinesterase and present interesting binding modes. Therefore, science has focused for the last five years on novel structures and their binding modes to acetylcholinesterase, leaving behind known drug structures that have been studied until now like tacrine, donepezil, or galantamine. It appeared that many studies are based on the comparison of the new drugs with the already existing drugs for better molecular docking analysis. From this study, it can be drawn which are the main important physicochemical characteristics for the interaction with CAS and PAS depending on the compound classes. Additionally, it seems that a wide variety of compound classes can be used as lead compounds for the design of novel, promising drugs for Alzheimer’s disease.

## Data Availability

Not applicable.

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
