# Peer review of "A Review on Recent Approaches on Molecular Docking Studies of Novel Compounds Targeting Acetylcholinesterase in Alzheimer Disease"

_molecules, 2023, doi:10.3390/molecules28031084_

Round 1
Reviewer 1 Report
In this manuscript, the authors outline recent progress of molecular docking studies of compounds targeting acetylcholinesterase (AChE) involved in Alzheimer’s disease. This manuscript includes comprehensive information regarding AChE inhibitors, providing valuable insights into Alzheimer’s disease drug development targeting AChE. However, figure presentation of protein structure model and compound quotation in the main text is not adequate, and thus quite difficult to read. This reviewer would recommend this manuscript for publication in Molecules if the extensive revision could be adequately performed.
Major points:
Lines 42-61: It is quite difficult to understand the structural information of the active site of AChE without the figure presentation, showing the catalytic triad and surrounding amino acid residues together with PAS and CAS positions. The authors should include the additional figure.
Line 83 and others: Like line 421, the authors should include quotations of figure and compound in the main text.
Figures 1-4, 10-12, 15-28, and 31-32: It is also quite difficult to understand the binding mode between compounds and AChE without the figure presentation of protein structure. Like Figure 13-14 and 29-30, the authors should include the additional figure throughout the manuscript (at least in the supplemental materials).
Minor points:
Line 38: What is type-B? The category information of other type(s) would be informative.
Line 112: nitrogen should read as amide.
Line 161: pai-interaction should read pai-pai-interaction
Lines 195-217: Figure 6 and 7 legend are redundant with Figure 5. It should be written in more abbreviated form.
Figures 5-9 (D & E): Schematic cartoon of the docking mode is very difficult to see due to the low resolution.
Figure 13-14 and 29-30: Figures 13 (29) and 14 (30) should be combined into one figure.
Author Response
In this manuscript, the authors outline recent progress of molecular docking studies of compounds targeting acetylcholinesterase (AChE) involved in Alzheimer’s disease. This manuscript includes comprehensive information regarding AChE inhibitors, providing valuable insights into Alzheimer’s disease drug development targeting AChE. However, figure presentation of protein structure model and compound quotation in the main text is not adequate, and thus quite difficult to read. This reviewer would recommend this manuscript for publication in Molecules if the extensive revision could be adequately performed.
We have added a new scheme defining the protein structure model and compound quotation in the main text to help the reader.
Major points:
Lines 42-61: It is quite difficult to understand the structural information of the active site of AChE without the figure presentation, showing the catalytic triad and surrounding amino acid residues together with PAS and CAS positions. The authors should include the additional figure.
A new schematic figure highlighting the active center of AChE depicting the catalytic triad and surrounding amino acid residues together with PAS and CAS positions has been added at the introduction section providing more structural information.
Line 83 and others: Like line 421, the authors should include quotations of figure and compound in the main text.
Thank you for this suggestion by mistake we have overlooked some quotations of figure and compound in the main text. We have corrected this, and all the quotations have been added in the main text.
Figures 1-4, 10-12, 15-28, and 31-32: It is also quite difficult to understand the binding mode between compounds and AChE without the figure presentation of protein structure. Like Figure 13-14 and 29-30, the authors should include the additional figure throughout the manuscript (at least in the supplemental materials).
It is true that Figures 13-14 and 29-30 clarify in detail the binding mode between compounds and AChE. We wanted to include figures for the binding interactions of all the novel derivatives (Figures 1-4, 10-12, 15-28, and 31-32) but there were under copyright. We have applied for a copyright permission, but still we have not received any yet, that is why we have not added the additional figures in our manuscript.
Minor points:
Line 38: What is type-B? The category information of other type(s) would be informative.
An explanation has been added in the manuscript.
Line 112: nitrogen should read as amide.
It has been corrected.
Line 161: pai-interaction should read pai-pai-interaction
It has been corrected.
Lines 195-217: Figure 6 and 7 legend are redundant with Figure 5. It should be written in more abbreviated form.
The Figures 5, 6, 7 present the interactions of different compounds 8, 9, 10 with the enzyme. We have kept the legends as they are in the original paper so as to be clear to the reader the differences in the binding mode of the different derivatives.
Figures 5-9 (D & E): Schematic cartoon of the docking mode is very difficult to see due to the low resolution.
All the figures have been added from the beginning and their resolution is the one in the original articles.
Figure 13-14 and 29-30: Figures 13 (29) and 14 (30) should be combined into one figure.
These figures present different compounds interactions that is why we have not combined them into one figure.
Reviewer 2 Report
The authors have collected quite a few AChE inhibitors, but the information included in this review does not correspond to the title of this review, nor deliver enough merits in general.
Please add all the information/discussions listed below to increase the overall merit of this review:
-
Comprehensive inhibition activity values should list explicitly in a table to help readers to have an overview of the number, and type of known AChE inhibitors.
-
The docking study is not comprehensive at all, the docking images are in low resolution. Please perform docking on all AChE inhibitors included in this review, and update the docking images.
-
The authors also mentioned there are three binding sites in the introduction, please categorise all AChE inhibitors based on your docking results accordingly.
-
ADME, and PK/PD profiles of AChE inhibitors are important information and should be covered and discussed in the review.
-
The AChE inhibitors with low potency and without potency would also offer valuable information for readers who are interested in designing novel AChE inhibitors. Please include these compounds as well.
Author Response
The authors have collected quite a few AChE inhibitors, but the information included in this review does not correspond to the title of this review, nor deliver enough merits in general.
Firstly, we would like to thank the reviewer for the fruitful comments. In this review data from the literature of the last five years (2018-2022) of novel AChE inhibitors have been collected. We aim to highlight the interaction of the designed and synthesized novel derivatives with AChE based on molecular modeling and docking and simulation techniques. We have not collected data before 2018 because there is another review covering the previous years (Biomolecules 2021, 11, 580. https://doi.org/10.3390/biom11040580).
Please add all the information/discussions listed below to increase the overall merit of this review:
- Comprehensive inhibition activity values should list explicitly in a table to help readers to have an overview of the number, and type of known AChE inhibitors.
A Table (Table I) has been added at the supplementary material listing the compounds, the used reference compounds and their inhibitory values as the reviewer suggested.
- The docking study is not comprehensive at all, the docking images are in low resolution. Please perform docking on all AChE inhibitors included in this review and update the docking images.
All the included docking images have been added from the beginning and their resolution is the one in the original articles.
Unfortunately, we are not able to perform docking studies on all AChE inhibitors included in this review and update the docking images since it is a review paper and not research one, so we have collected data from the literature. Additionally, we are not able to perform docking studies on all AChE inhibitors because we have no licenses of the different softwares used in the original research articles.
As for the additional images not included in this review, we have applied for a copyright permission, but we have not received any yet, that is why we have not added the additional figures in our manuscript.
- The authors also mentioned there are three binding sites in the introduction, please categorise all AChE inhibitors based on your docking results accordingly.
The AChE inhibitors retrieved from the literature and referred in this review are categorized based on the docking results accordingly in supplementary material in Table I.
- ADME, and PK/PD profiles of AChE inhibitors are important information and should be covered and discussed in the review.
This review is focused on docking studies of novel AChE inhibitors covering data from recent literature. ADME and PK/PD profiles of AChE inhibitors are important information, but this is not the topic of this review. Additionally there is no ADME, and PK/PD data for all the novel derivatives from the literature mentioned in this review that is why we have not added them.
- The AChE inhibitors with low potency and without potency would also offer valuable information for readers who are interested in designing novel AChE inhibitors. Please include these compounds as well.
We agree with the reviewer’s comment but unfortunately, the binding mode of the inactive compounds is not studied even at the original papers that is why they are not included in this review.
Reviewer 3 Report
What is the novelty of this paper. Also images are taken from other sources. Introduction need improvement. Molecular mechanisms of Alzheimer's should be added.
Author Response
Comments and Suggestions for Authors
What is the novelty of this paper. Also, images are taken from other sources. Introduction need improvement. Molecular mechanisms of Alzheimer's should be added.
What is the novelty of this paper.
This paper discusses the structures, the binding mode to AChE of novel inhibitors as well as the physicochemical requirements for the design of potential AChE inhibitors from 2018-till now. The rest of the data before 2018 are covered from a different review (Biomolecules 2021, 11, 580. https://doi.org/10.3390/biom11040580) as mentioned in the manuscript.
Also, images are taken from other sources.
It is true that certain images highlighting the binding mode are taken from other sources (no copyright permission required) but are important because they provide the reader valuable information for the binding mode of the novel derivatives.
Introduction need improvement.
The introduction has been written from the beginning.
Molecular mechanisms of Alzheimer's should be added.
The molecular mechanisms of Alzheimer's have been added and explained in a detailed manner in the introduction.
Round 2
Reviewer 1 Report
The authors performed an extensive revision, and adequately responded to this reviewer's comments.